# Generalization Bounds for Uniformly Stable Algorithms

**Vitaly Feldman**
Google Brain

**Jan Vondrak**
Stanford University

## Abstract

Uniform stability of a learning algorithm is a classical notion of algorithmic stability introduced to derive high-probability bounds on the generalization error (Bousquet and Elisseeff, 2002). Specifically, for a loss function with range bounded in $[0, 1]$, the generalization error of a $\gamma$-uniformly stable learning algorithm on $n$ samples is known to be within $O((\gamma + 1/n)\sqrt{n \log(1/\delta)})$ of the empirical error with probability at least $1 - \delta$. Unfortunately, this bound does not lead to meaningful generalization bounds in many common settings where $\gamma \geq 1/\sqrt{n}$. At the same time the bound is known to be tight only when $\gamma = O(1/n)$.

We substantially improve generalization bounds for uniformly stable algorithms without making any additional assumptions. First, we show that the bound in this setting is $O(\sqrt{(\gamma + 1/n) \log(1/\delta)})$ with probability at least $1 - \delta$. In addition, we prove a tight bound of $O(\gamma^2 + 1/n)$ on the second moment of the estimation error. The best previous bound on the second moment is $O(\gamma + 1/n)$. Our proofs are based on new analysis techniques and our results imply substantially stronger generalization guarantees for several well-studied algorithms.

## 1 Introduction

We consider the basic problem of estimating the generalization error of learning algorithms. Over the last couple of decades, a remarkably rich and deep theory has been developed for bounding the generalization error via notions of complexity of the class of models (or predictors) output by the learning algorithm. At the same time, for a variety of learning algorithms this theory does not provide satisfactory bounds (even as compared with other theoretical analyses). Most notable among these are continuous optimization algorithms that play the central role in modern machine learning. For example, the standard generalization error bounds for stochastic gradient descent (SGD) on convex Lipschitz functions cannot be obtained by proving uniform convergence for all empirical risk minimizers (ERM) [13, 26]. Specifically, there exist empirical risk minimizing algorithms whose generalization error is $\sqrt{d}$ times larger than the generalization error of SGD, where $d$ is the dimension of the problem (without the Lipschitzness assumption the gap is infinite even for $d = 2$) [13]. This disparity stems from the fact that uniform convergence bounds largely ignore the way in which the model output by the algorithm depends on the data. We note that in the restricted setting of generalized linear models one can obtain tight generalization bounds via uniform convergence [15].

Another classical and popular approach to proving generalization bounds is to analyze the stability of the learning algorithm to changes in the dataset. This approach has been used to obtain relatively strong generalization bounds for several convex optimization algorithms. For example, the seminal works of Bousquet and Elisseeff [4] and Shalev-Shwartz et al. [26] demonstrate that for strongly convex losses the ERM solution is stable. The use of stability is also implicit in standard analyses of online convex optimization [26] and online-to-batch conversion [5]. More recently, Hardt et al. [14] showed that for convex smooth losses the solution obtained via (stochastic) gradient descent is

stable. They also conjectured that stability can be used to understand the generalization properties of algorithms used for training deep neural networks.

While a variety of notions of stability have been proposed and analyzed, most only lead to bounds on the expectation or the second moment of the estimation error over the random choice of the dataset (where estimation error refers to the difference between the true generalization error and the empirical error). In contrast, generalization bounds based on uniform convergence show that the estimation error is small with high probability (more formally, the distribution of the error has exponentially decaying tails). This discrepancy was first addressed by Bousquet and Elisseeff [4] who defined the notion of *uniform stability*.

**Definition 1.1.** *Let $A\colon Z^n \to \mathcal{F}$ be a learning algorithm mapping a dataset $S$ to a model in $\mathcal{F}$ and $\ell\colon \mathcal{F} \times Z \to \mathbb{R}$ be a function such that $\ell(f, z)$ measures the loss of model $f$ on point $z$. Then $A$ is said to have uniform stability $\gamma_n$ with respect to $\ell$ if for any pair of datasets $S, S' \in Z^n$ that differ in a single element and every $z \in Z$, $|\ell(A(S), z) - \ell(A(S'), z)| \leq \gamma_n$.*

We denote the empirical loss of the algorithm $A$ on $S = (S_1, \ldots, S_n)$ by $\mathcal{E}_S[\ell(A(S))] \doteq \frac{1}{n} \sum_{i=1}^{n} \ell(A(S), S_i)$ and its expected loss relative to distribution $\mathcal{P}$ over $Z$ by $\mathcal{E}_{\mathcal{P}}[\ell(A(S))] \doteq \mathbf{E}_{z \sim \mathcal{P}}[\ell(A(S), z)]$. We denote the estimation error[1] of $A$ on $S$ relative to $\mathcal{P}$ by

$$\Delta_{\mathcal{P}-S}(\ell(A)) \doteq \mathcal{E}_{\mathcal{P}}[\ell(A(S))] - \mathcal{E}_S[\ell(A(S))].$$

We summarize the generalization properties of uniform stability in the below (all proved in [4] although properties (1) and (2) are implicit in earlier work and also hold under weaker stability notions). Let $A\colon Z^n \to \mathcal{F}$ be a learning algorithm that has uniform stability $\gamma_n$ with respect to a loss function $\ell : \mathcal{F} \times Z \to [0, 1]$. Then for every distribution $\mathcal{P}$ over $Z$ and $\delta > 0$:

$$\left| \mathop{\mathbf{E}}_{S \sim \mathcal{P}^n} [\Delta_{\mathcal{P}-S}(\ell(A))] \right| \leq \gamma_n; \tag{1}$$

$$\mathop{\mathbf{E}}_{S \sim \mathcal{P}^n} \left[ (\Delta_{\mathcal{P}-S}(\ell(A)))^2 \right] \leq \frac{1}{2n} + 6\gamma_n; \tag{2}$$

$$\mathop{\mathbf{Pr}}_{S \sim \mathcal{P}^n} \left[ \Delta_{\mathcal{P}-S}(\ell(A)) \geq \left( 4\gamma_n + \frac{1}{n} \right) \sqrt{\frac{n \ln(1/\delta)}{2}} + 2\gamma_n \right] \leq \delta. \tag{3}$$

As can be readily seen from eq.(3) the high probability bound is at least a factor $\sqrt{n}$ larger than the expectation of the estimation error. In addition, the bound on the estimation error implied by eq.(2) is quadratically worse than the stability parameter. We note that eq. (1) does not imply that $\mathcal{E}_{\mathcal{P}}[\ell(A(S))] \leq \mathcal{E}_S[\ell(A(S))] + O(\gamma_n/\delta)$ with probability at least $1 - \delta$ since $\Delta_{\mathcal{P}-S}(\ell(A))$ can be negative and Markov's inequality cannot be used. Such "low-probability" result is known only for ERM algorithms for which Shalev-Shwartz et al. [26] showed that

$$\mathop{\mathbf{E}}_{S \sim \mathcal{P}^n} [|\Delta_{\mathcal{P}-S}(\ell(A))|] \leq O \left( \gamma_n + \frac{1}{\sqrt{n}} \right) \tag{4}$$

Naturally, for most algorithms the stability parameter needs be balanced against the guarantees on the empirical error. For example, ERM solution to convex learning problems can be made uniformly stable by adding a strongly convex term to the objective [26]. This change in the objective introduces an error. In the other example, the stability parameter of gradient descent on smooth objectives is determined by the sum of the rates used for all the gradient steps [14]. Limiting the sum limits the empirical error that can be achieved. In both of those examples the optimal expected error can only be achieved when $\gamma_n = \theta(1/\sqrt{n})$ (which is also the expected suboptimality of the solutions). Unfortunately, in this setting, eq. (3) gives a vacuous bound and only "low-probability" generalization bounds are known for the first example (since it is ERM and eq. (4) applies).

This raises a natural question of whether the known bounds in eq. (2) and eq. (3) are optimal. In particular, Shalev-Shwartz et al. [26] conjecture that better high probability bounds can be achieved.

It is easy to see that the expectation of the absolute value of the estimation error can be at least $\gamma_n + \frac{1}{\sqrt{n}}$. Consequently, as observed already in [4], eq. (3) is optimal when $\gamma_n = O(1/n)$. (Note that this is the optimal level of stability for non-trivial learning algorithms with $\ell$ normalized to $[0, 1]$.) Yet both bounds in eq. (2) and eq.(3) are significantly larger than this lower bound whenever $\gamma_n = \omega(1/n)$. At the same time, to the best of our knowledge, no other upper or lower bounds on the estimation error of uniformly stable algorithms were previously known.

## 1.1 Our Results

We give two new upper bounds on the estimation error of uniformly stable learning algorithms. Specifically, our bound on the second moment of the estimation error is $O(\gamma_n^2 + 1/n)$ matching (up to a constant) the simple lower bound of $\gamma_n + \frac{1}{\sqrt{n}}$ on the first moment. Our high probability bound improves the rate from $\sqrt{n}(\gamma_n + 1/n)$ to $\sqrt{\gamma_n + 1/n}$. This rate is non-vacuous for any non-trivial stability parameter $\gamma_n = o(1)$ and matches the rate that was previously known only for the second moment (eq. (2)).

For convenience and generality we state our bounds on the estimation error for arbitrary data dependent functions (and not just losses of models). Specifically, let $M : Z^n \times Z \to \mathbb{R}$ be an algorithm that is given a dataset $S$ and a point $z$ as an input. It can be thought of as computing a real-valued function $M(S, \cdot)$ and then applying it to $z$. In the case of learning algorithms $M(S, z) = \ell(A(S), z)$ but this notion also captures other data statistics whose choice may depend on the data. We denote the empirical mean $\mathcal{E}_S[M(S)] \doteq \frac{1}{n} \sum_{i=1}^n M(S, S_i)$, expectation relative to distribution $\mathcal{P}$ over $Z$ by $\mathcal{E}_{\mathcal{P}}[M(S)] \doteq \mathbf{E}_{z \sim \mathcal{P}}[M(S, z)]$ and the estimation error by

$$\Delta_{\mathcal{P}-S}(M) \doteq \mathcal{E}_{\mathcal{P}}[M(S)] - \mathcal{E}_S[M(S)].$$

Uniform stability for data-dependent functions is defined analogously (Def. 2.1).

**Theorem 1.2.** *Let $M : Z^n \times Z \to [0, 1]$ be a data-dependent function with uniform stability $\gamma_n$. Then for any probability distribution $\mathcal{P}$ over $Z$ and any $\delta \in (0, 1)$:*

$$\mathbf{E}_{S \sim \mathcal{P}^n} \left[ (\Delta_{\mathcal{P}-S}(M))^2 \right] \leq 16\gamma_n^2 + \frac{2}{n}; \tag{5}$$

$$\mathbf{Pr}_{S \sim \mathcal{P}^n} \left[ \Delta_{\mathcal{P}-S}(M) \geq 8\sqrt{\left(2\gamma_n + \frac{1}{n}\right) \cdot \ln(8/\delta)} \right] \leq \delta. \tag{6}$$

The results in Theorem 1.2 are stated only for deterministic functions (or algorithms). They can be extended to randomized algorithms in several standard ways [12, 26]. If $M$ is uniformly $\gamma$-stable with high probability over the choice of its random bits then one can obtain a statement which holds with high probability over the choice of both $S$ and the random bits (e.g. [19]). Alternatively, one can always consider the function $M'(S, z) = \mathbf{E}_M[M(S, z)]$. If $M'(S, z)$ is uniformly $\gamma$-stable then Thm. 1.2 can be applied to it. The resulting statement will be only about the expected value of the estimation error with expectation taken over the randomness of the algorithm. Further, if $M$ is used with independent randomness in each evaluation of $M(S, S_i)$ then the empirical mean $\mathcal{E}_S[M(S)]$ will be strongly concentrated around $\mathcal{E}_S[M'(S)]$ (whenever the variance of each evaluation is not too large). We note that randomized algorithms also allow to extend the notion of uniform stability to binary classification algorithms by considering the expectation of the $0/1$ loss.

A natural and, we believe, important question left open by our work is whether the high probability result in eq. (6) is tight.

**Our techniques** The high-probability generalization result in [4] (eq. (3)) is based on a simple observation that as a function of $S$, $\Delta_{\mathcal{P}-S}(M)$ has the bounded differences property. Replacing any element of $S$ can change $\Delta_{\mathcal{P}-S}(M)$ by at most $2\gamma_n + 1/n$ (where $\gamma_n$ comes from changing the function $M(S, \cdot)$ to $M(S', \cdot)$ and $1/n$ comes the change in one of the points on which this function is evaluated). Applying McDiarmid's concentration inequality immediately implies concentration with rate $\sqrt{n}(2\gamma_n + 1/n)$ around the expectation. The expectation, in turn, is small by eq. (1). In contrast, our approach uses stability itself as a tool for proving concentration inequalities. It is based on ideas developed in [2] to prove generalization bounds for differentially private algorithms in the

context of adaptive data analysis [11]. It was recently shown that this proof approach can be used to re-derive and extend several standard concentration inequalities [23, 27].

At a high level, the first step of the argument reduces the task of proving a bound on the tail of a non-negative real-valued random variable to bounding the expectation of the maximum of multiple independent samples of that random variable. We then show that from multiple executions of $M$ on independently chosen datasets it is possible to select the execution of $M$ with approximately the largest estimation error in a stable way. That is, uniform stability of $M$ allows us to ensure that the selection procedure is itself uniformly stable. The selection procedure is based on the exponential mechanism [21] and satisfies differential privacy [9](Def. 2.3). The stability of this procedure allows us to bound the expectation of the estimation error of the execution of $M$ with approximately the largest estimation error (among the multiple executions). This gives us the desired bound on the expectation of the maximum of multiple independent samples of the estimation error random variable. We remark that the multiple executions and an algorithm for selecting among them exist purely for the purposes of the proof technique and do not require any modifications to the algorithm itself.

Our approach to proving the bound on the second moment of the estimation error is based on two ideas. First we decouple the point on which each $M(S)$ is estimated from $S$ by observing that for every dataset $S$ the empirical mean is within $2\gamma_n$ of the "leave-one-out" estimate of the true mean. Specifically, our leave-one-out estimator is defined as $\mathbf{E}_{z\sim\mathcal{P}}\left[\frac{1}{n}\sum_{i=1}^{n}M(S^{i\leftarrow z},S_i)\right]$, where $S^{i\leftarrow z}$ denotes replacing the element in $S$ at index $i$ with $z$. We then bound the second moment of the estimation error of the leave-one-out estimate by bounding the effect of dependence between the random variables by $O(\gamma_n^2+1/n)$.

**Applications**  We now apply our bounds on the estimation error to several known uniformly stable algorithms in a straightforward way. Our main focus are learning problems that can be formulated as stochastic convex optimization. Specifically, these are problems in which the goal is to minimize the expected loss: $F_{\mathcal{P}}(w)\doteq\mathbf{E}_{z\sim\mathcal{P}}[\ell(w,z)]$ over $w\in\mathcal{K}\subset\mathbb{R}^d$ for some convex body $\mathcal{K}$ and a family of convex losses $\mathcal{F}=\{\ell(\cdot,z)\}_{z\in Z}$. The stochastic convex optimization problem for a family of losses $\mathcal{F}$ over $\mathcal{K}$ is the problem of minimizing $F_{\mathcal{P}}(w)$ for an arbitrary distribution $\mathcal{P}$ over $Z$.

For concreteness, we consider the well-studied setting in which $\mathcal{F}$ contains 1-Lipschitz convex functions with range in $[0,1]$ and $\mathcal{K}$ is included in the unit ball. In this case ERM with a strongly convex regularizer $\frac{\lambda}{2}\|w\|^2$ has uniform stability of $1/(\lambda n)$ [4, 26]. From here, applying Markov's inequality to eq. (4), Shalev-Shwartz et al. [26] obtain a "low-probability" generalization bound for the solution. Their bound on the true loss is within $O(1/\sqrt{\delta n})$ from the optimum with probability at least $1-\delta$. Applying eq. (5) with Chebyshev's inequality improves the dependence on $\delta$ quadratically, that is to $O(1/(\delta^{1/4}\sqrt{n}))$. Further, using eq. (5) we obtain that for an appropriate choice of $\lambda$, the sub-optimality of the solution is at most $O(\sqrt{\log(1/\delta)}/n^{1/3})$.

Another algorithm that was shown to be uniformly stable is gradient descent[2] on sufficiently smooth convex functions [14]. The generalization bounds we obtain for this algorithm are similar to those we get for the strongly convex ERM. We note that for the stability-based analysis in this case even "low-probability" generalization bounds were not known for the optimal error rate of $1/\sqrt{n}$.

Finally, we show that our results can be used to improve the recent bounds on estimation error of learning algorithms with differentially private prediction. These are algorithms introduced to model privacy-preserving learning in the settings where users only have black-box access to the learned model via a prediction interface [10]. The properties of differential privacy imply that the expectation over the randomness of a predictor $K\colon(X\times Y)^n\times X$ of the loss of $K$ at any point $x\in X$ is uniformly stable. Specifically, for an $\epsilon$-differentially private prediction algorithm, every loss function $\ell\colon Y\times Y\to[0,1]$, two datasets $S,S'\in(X\times Y)^n$ that differ in a single element and $(x,y)\in X\times Y$:

$$\left|\mathbf{E}_{K}[\ell(K(S,x),y)]-\mathbf{E}_{M}[\ell(K(S',x),y)]\right|\le e^{\epsilon}-1.$$

Therefore, our generalization bounds can be directly applied to the data-dependent function $M(S,(x,y))\doteq\mathbf{E}_K[\ell(K(S,x),y)]$. These bounds can, in turn, be used to get stronger genera-

lization bounds for one of the learning algorithms proposed in [10] (that has unbounded model complexity).

Additional details of these applications can be found in the supplemental material.

## 1.2 Additional related work

The use of stability for understanding of generalization properties of learning algorithms dates back to the pioneering work of Rogers and Wagner [25]. They showed that expected sensitivity of a classification algorithm to changes of individual examples can be used to obtain a bound on the variance of the leave-one-out estimator for the $k$-NN algorithm. Early work on stability focused on extensions of these results to other "local" algorithms and estimators and focused primarily on variance (a notable exception is [8] where high probability bounds on the generalization error of $k$-NN are proved). See [7] for an overview. In a somewhat similar spirit, stability is also used for analysis of the variance of the $k$-fold cross-validation estimator [3, 16, 17].

A long line of work focuses on the relationship between various notions of stability and learnability in supervised setting (see [24, 26] for an overview). This work employs relatively weak notions of average stability and derives a variety of asymptotic equivalence results. The results in [4] on uniform stability and their applications to generalization properties of strongly convex ERM algorithms have been extended and generalized in several directions (e.g. [18, 28, 30]). Maurer [20] considers generalization bounds for a special case of linear regression with a strongly convex regularizer and a sufficiently smooth loss function. Their bounds are data-dependent and are potentially stronger for large values of the regularization parameter (and hence stability). However the bound is vacuous when the stability parameter is larger than $n^{-1/4}$ and hence is not directly comparable to ours. Finally, recent work of Abou-Moustafa and Szepesvári [1] gives high-probability generalization bounds similar to those in [4] but using a bound on a high-order moment of stability instead of the uniform stability. We also remark that all these works are based on techniques different from ours.

Uniform stability plays an important role in privacy-preserving learning since a differentially private learning algorithm can usually be obtained one by adding noise to the output of a uniformly stable one (e.g. [6, 10, 29]).

## 2 Preliminaries

For a domain $Z$, a dataset $S \in Z^n$ is an $n$-tuple of elements in $Z$. We refer to element with index $i$ by $S_i$ and by $S^{i \leftarrow z}$ to the dataset obtained from $S$ by setting the element with index $i$ to $z$. We refer to a function that takes as an input a dataset $S \in Z^n$ and a point $z \in Z$ as a *data-dependent function* over $Z$. We think of data-dependent functions as outputs of an algorithm that takes $S$ as an input. For example in supervised learning $Z$ is the set of all possible labeled examples $Z = X \times Y$ and the algorithm $M$ is defined as estimating some loss function $\ell_Y : Y \times Y \to \mathbb{R}_+$ of the model $h_S$ output by a learning algorithm $A(S)$ on example $z = (x, y)$. That is $M(S, z) = \ell_Y(h_S(x), y)$. Note that in this setting $\mathcal{E}_{\mathcal{P}}[M(S)]$ is exactly the true loss of $h_S$ on data distribution $\mathcal{P}$, whereas $\mathcal{E}_S[M(S)]$ is the empirical loss of $h_S$.

**Definition 2.1.** *A data-dependent function $M : Z^n \times Z \to \mathbb{R}$ has uniform stability $\gamma$ if for all $S \in Z^n$, $i \in [n]$, $z_i, z \in Z$, $|M(S, z) - M(S^{i \leftarrow z_i}, z)| \leq \gamma$.*

This definition is equivalent to having $M(S, z)$ having *sensitivity* $\gamma$ or $\gamma$-bounded differences for all $z \in Z$.

**Definition 2.2.** *A real-valued function $f : Z^n \to \mathbb{R}$ has sensitivity at most $\gamma$ if for all $S \in Z^n$, $i \in [n]$, $z_i, z \in Z$, $|f(S) - f(S^{i \leftarrow z_i})| \leq \gamma$.*

We will also rely on several elementary properties of differential privacy [9]. In this context differential privacy is simply a form of uniform stability for randomized algorithms.

**Definition 2.3** ([9]). *An algorithm $A : Z^n \to Y$ is $\epsilon$-differentially private if, for all datasets $S, S' \in Z^n$ that differ on a single element,*

$$\forall E \subseteq Y \quad \mathbf{Pr}[A(S) \in E] \leq e^\epsilon \mathbf{Pr}[A(S') \in E].$$

# 3  Generalization with Exponential Tails

Our approach to proving the high-probability generalization bounds is based on the technique introduced by Nissim and Stemmer [22] (see [2]) to show that differentially private algorithm have strong generalization properties. It has recently been pointed out by Steinke and Ullman [27] that this approach can be used to re-derive the standard Bernstein, Hoeffding, and Chernoff concentration inequalities. Nissim and Stemmer [23] used the same approach to generalize McDiarmid's inequality to functions with unbounded (or high) sensitivity.

We prove a bound on the tail of a random variable by bounding the expectation of the maximum of multiple independent samples of the random variable. Specifically, the following simple lemma (see [27] for proof):

**Lemma 3.1.** *Let $\mathcal{Q}$ be a probability distribution over the reals. Then*

$$\Pr_{v \sim \mathcal{Q}}\left[v \geq 2 \cdot \mathop{\mathbf{E}}_{v_1, \ldots, v_m \sim \mathcal{Q}}[\max\{0, v_1, v_2, \ldots, v_m\}]\right] \leq \frac{\ln(2)}{m}.$$

The second step relies on the relationship between the maximum of a set of values and the value chosen by the soft-argmax, which we refer to as the *stable-max*. Specifically, we define

$$\text{stablemax}_\epsilon\{v_1, \ldots, v_m\} \doteq \sum_{i \in [m]} v_i \cdot \frac{e^{\epsilon v_i}}{\sum_{\ell \in [m]} e^{\epsilon v_\ell}},$$

where $\frac{e^{\epsilon v_i}}{\sum_{\ell \in [m]} e^{\epsilon v_\ell}}$ should be thought of as the relative weight assigned to value $v_i$. (We remark that this vector of weights is commonly referred to as softmax and soft-argmax. We therefore use stable-max to avoid confusion between the weights and the weighted sum of values.) The first property of the stable-max is that its value is close to the maximum:

$$\text{stablemax}_\epsilon\{v_1, \ldots, v_m\} \geq \max\{v_1, \ldots, v_m\} - \frac{\ln m}{\epsilon}.$$

The second property that we will use is that the weight (or probability) assigned to each value is stable: it changes by a factor of at most $e^{2\gamma\epsilon}$ whenever each of the values changes by at most $\gamma$. These two properties are known properties of the exponential mechanism [21]. More formally, the exponential mechanism is the randomized algorithm that given values $\{v_1, \ldots, v_m\}$ and $\epsilon$, outputs the index $i$ with probability $\frac{e^{\epsilon v_i}}{\sum_{\ell \in [m]} e^{\epsilon v_\ell}}$. We state the properties of the exponential mechanism specialized to our context below.

**Theorem 3.2.** *[2, 21] Let $f_1, \ldots, f_m : Z^n \to \mathbb{R}$ be $m$ scoring functions of a dataset each of sensitivity at most $\Delta$. Let $A$ be the algorithm that given a dataset $S \in Z^n$ and a parameter $\epsilon > 0$ outputs an index $\ell \in [m]$ with probability proportional to $e^{\frac{\epsilon}{2\Delta} \cdot f_\ell(S)}$. Then $A$ is $\epsilon$-differentially private and, further, for every $S \in Z^n$:*

$$\mathop{\mathbf{E}}_{\ell = A(S)}[f_\ell(S)] \geq \max_{\ell \in [m]}\{f_\ell(S)\} - \frac{2\Delta}{\epsilon} \cdot \ln m.$$

We now define the scoring functions designed to select the execution of $M$ with the worst estimation error. For these purposes our dataset will consist of $m$ datasets each of size $n$. To avoid confusion, we emphasize this by referring to it as multi-dataset and using $\mathcal{S}$ to denote it. That is $\mathcal{S} \in Z^{m \times n}$ and we refer to each of the sub-datasets as $\mathcal{S}_1, \ldots, \mathcal{S}_m$ and to an element $i$ of sub-dataset $\ell$ as $\mathcal{S}_{\ell,i}$.

**Lemma 3.3.** *Let $M : Z^n \times Z \to [0, 1]$ be a data-dependent function with uniform stability $\gamma$. For a probability distribution $\mathcal{P}$ over $Z$, multi-dataset $\mathcal{S} \in Z^{m \times n}$ and an index $\ell \in [m]$ we define the scoring function*

$$f_\ell(\mathcal{S}) \doteq \Delta_{\mathcal{P} - \mathcal{S}_\ell}(M) = \mathcal{E}_{\mathcal{P}}[M(\mathcal{S}_\ell)] - \mathcal{E}_{\mathcal{S}_\ell}[M(\mathcal{S}_\ell)].$$

*Then $f_\ell$ has sensitivity $2\gamma + 1/n$.*

*Proof.* Let $\mathcal{S}$ and $\mathcal{S}'$ be two multi-datasets that differ in a single element at index $i$ in sub-dataset $k$. Clearly, if $k \neq \ell$ then $\mathcal{S}_\ell = \mathcal{S}'_\ell$ and $f_\ell(\mathcal{S}) = f_\ell(\mathcal{S}')$. Otherwise, $\mathcal{S}_\ell$ and $\mathcal{S}'_\ell$ differ in a single element. Thus

$$|\mathcal{E}_{\mathcal{P}}[M(\mathcal{S}_\ell)] - \mathcal{E}_{\mathcal{P}}[M(\mathcal{S}'_\ell)]| = \left|\mathop{\mathbf{E}}_{z \sim \mathcal{P}}[M(\mathcal{S}_\ell, z) - M(\mathcal{S}'_\ell, z)]\right| \leq \gamma.$$

and

$$\left| \mathcal{E}_{\mathcal{S}_\ell}[M(\mathcal{S}_\ell)] - \mathcal{E}_{\mathcal{S}'_\ell}[M(\mathcal{S}'_\ell)] \right| = \left| \frac{1}{n} \sum_{j \in [n]} M(\mathcal{S}_\ell, \mathcal{S}_{\ell,j}) - \frac{1}{n} \sum_{j \in [n]} M(\mathcal{S}'_\ell, \mathcal{S}'_{\ell,j}) \right|$$

$$\leq \left| \frac{1}{n} \sum_{j \in [n], j \neq i} (M(\mathcal{S}_\ell, \mathcal{S}_{\ell,j}) - M(\mathcal{S}'_\ell, \mathcal{S}_{\ell,j})) \right| + \frac{1}{n} \cdot \left| M(\mathcal{S}'_\ell, \mathcal{S}_{\ell,i}) - M(\mathcal{S}'_\ell, \mathcal{S}'_{\ell,i}) \right|$$

$$\leq \gamma + \frac{1}{n}.$$

$\square$

The final (and new) ingredient of our proof is a bound on the expected estimation error of any uniformly stable algorithm on a sub-dataset chosen in a differentially private way.

**Lemma 3.4.** *For $\ell \in [m]$, let $M_\ell : Z^n \times Z \to [0,1]$ be a data-dependent function with uniform stability $\gamma$. Let $A : Z^{n \times m} \to [m]$ be an $\epsilon$-differentially private algorithm. Then for any distribution $\mathcal{P}$ over $Z$, we have that:*

$$e^{-\epsilon} V_{\mathcal{S}} - \gamma \leq \mathop{\mathbf{E}}_{\mathcal{S} \sim \mathcal{P}^{mn}, \ell = A(\mathcal{S})} [\mathcal{E}_{\mathcal{P}}[M_\ell(\mathcal{S}_\ell)]] \leq e^\epsilon V_{\mathcal{S}} + \gamma,$$

*where $V_{\mathcal{S}} \doteq \mathbf{E}_{\mathcal{S} \sim \mathcal{P}^{mn}, \ell = A(\mathcal{S})} [\mathcal{E}_{\mathcal{S}_\ell}[M_\ell(\mathcal{S}_\ell)]].$*

*Proof.*

$$V_{\mathcal{S}} = \mathop{\mathbf{E}}_{\mathcal{S} \sim \mathcal{P}^{mn}, \ell = A(\mathcal{S})} \left[ \frac{1}{n} \sum_{i \in [n]} M_\ell(\mathcal{S}_\ell, \mathcal{S}_{\ell,i}) \right]$$

$$= \mathop{\mathbf{E}}_{A, \mathcal{S} \sim \mathcal{P}^{mn}} \left[ \frac{1}{n} \sum_{i \in [n]} \sum_{\ell \in [m]} \mathbb{1}(A(\mathcal{S}) = \ell) \cdot M_\ell(\mathcal{S}_\ell, \mathcal{S}_{\ell,i}) \right]$$

$$= \frac{1}{n} \sum_{i \in [n]} \sum_{\ell \in [m]} \mathop{\mathbf{E}}_{\mathcal{S} \sim \mathcal{P}^{mn}} \left[ \mathop{\mathbf{E}}_{A}[\mathbb{1}(A(\mathcal{S}) = \ell)] \cdot M_\ell(\mathcal{S}_\ell, \mathcal{S}_{\ell,i}) \right]$$

$$\leq \frac{1}{n} \sum_{i \in [n]} \sum_{\ell \in [m]} \mathop{\mathbf{E}}_{\mathcal{S} \sim \mathcal{P}^{mn}, z \sim \mathcal{P}} \left[ e^\epsilon \cdot \mathop{\mathbf{E}}_{A}[\mathbb{1}(A(\mathcal{S}^{\ell, i \leftarrow z}) = \ell)] \cdot (M_\ell(\mathcal{S}_\ell^{i \leftarrow z}, \mathcal{S}_{\ell,i}) + \gamma) \right]$$

$$= \frac{1}{n} \sum_{i \in [n]} \sum_{\ell \in [m]} \mathop{\mathbf{E}}_{\mathcal{S} \sim \mathcal{P}^{mn}, z \sim \mathcal{P}} \left[ e^\epsilon \cdot \mathop{\mathbf{E}}_{A}[\mathbb{1}(A(\mathcal{S}) = \ell)] \cdot (M_\ell(\mathcal{S}_\ell, z) + \gamma) \right]$$

$$= \mathop{\mathbf{E}}_{\mathcal{S} \sim \mathcal{P}^{mn}, z \sim \mathcal{P}, \ell = A(\mathcal{S})} [e^\epsilon \cdot (M_\ell(\mathcal{S}_\ell, z) + \gamma)] = e^\epsilon \cdot \left( \mathop{\mathbf{E}}_{\mathcal{S} \sim \mathcal{P}^{mn}, z \sim \mathcal{P}, \ell = A(\mathcal{S})} [M_\ell(\mathcal{S}_\ell, z)] + \gamma \right).$$

This gives the left hand side of the stated inequality. The right hand side is obtained analogously. $\square$

We are now ready to put the ingredients together to prove the claimed result:

*Proof of eq. (6) in Theorem 1.2.* We choose $m = \ln(2)/\delta$. Let $f_1, \ldots, f_m$ be the scoring functions defined in Lemma 3.3. Let $f_{m+1}(\mathcal{S}) \equiv 0$. Let $A$ be the execution of the exponential mechanism with $\Delta = 2\gamma + 1/n$ on scoring functions $f_1, \ldots, f_{m+1}$ and $\epsilon$ to be defined later. Note that this corresponds to the setting of Lemma 3.4 with $M_\ell \equiv M$ for all $\ell \in [m]$ and $M_{m+1} \equiv 0$. By Lemma 3.4 we have that

$$\mathop{\mathbf{E}}_{\mathcal{S} \sim \mathcal{P}^{(m+1)n}} \left[ \mathop{\mathbf{E}}_{\ell = A(\mathcal{S})} [f_\ell(\mathcal{S})] \right] = \mathop{\mathbf{E}}_{\mathcal{S} \sim \mathcal{P}^{(m+1)n}, \ell = A(\mathcal{S})} [\mathcal{E}_{\mathcal{P}}[M_\ell(\mathcal{S}_\ell)] - \mathcal{E}_{\mathcal{S}_\ell}[M_\ell(\mathcal{S}_\ell)]] \leq e^\epsilon - 1 + \gamma.$$

By Theorem 3.2

$$\operatorname*{\mathbf{E}}_{\mathcal{S}\sim\mathcal{P}^{mn}}\left[\max\left\{0, \max_{\ell\in[m]}\mathcal{E}_{\mathcal{P}}[M(\mathcal{S}_\ell)] - \mathcal{E}_{\mathcal{S}_\ell}[M(\mathcal{S}_\ell)]\right\}\right] = \operatorname*{\mathbf{E}}_{\mathcal{S}\sim\mathcal{P}^{mn}}\left[\max_{\ell\in[0.m]}f_\ell(\mathcal{S})\right]$$

$$\leq \operatorname*{\mathbf{E}}_{\mathcal{S}\sim\mathcal{P}^{mn}}\left[\operatorname*{\mathbf{E}}_{\ell=A(\mathcal{S})}[f_\ell(\mathcal{S})]\right] + \frac{2\Delta}{\epsilon}\ln(m+1) \leq e^\epsilon - 1 + \gamma + \frac{4\gamma + 2/n}{\epsilon}\ln(m+1).$$

To bound this expression we choose $\epsilon = \sqrt{\left(2\gamma + \frac{1}{n}\right) \cdot \ln(m+1)} = \sqrt{\left(2\gamma + \frac{1}{n}\right) \cdot \ln(e\ln(2)/\delta)}$. Our bound is at least $2\epsilon$ and hence holds trivially if $\epsilon \geq 1/2$. Otherwise $(e^\epsilon - 1) \leq 2\epsilon$ and we obtain the following bound on the expectation of the maximum.

$$4\sqrt{\left(2\gamma + \frac{1}{n}\right) \cdot \ln(e\ln(2)/\delta)} + \gamma \leq 4\sqrt{\left(2\gamma + \frac{1}{n}\right) \cdot \ln(8/\delta)}$$

where we used that $\gamma \leq \sqrt{\gamma}$. Finally, plugging this bound into Lemma 3.1 we obtain that

$$\operatorname*{\mathbf{Pr}}_{S\sim\mathcal{P}^n}\left[\mathcal{E}_{\mathcal{P}}[M(S)] - \mathcal{E}_S[M(S)] \geq 8\sqrt{\left(2\gamma + \frac{1}{n}\right) \cdot \ln(8/\delta)}\right] \leq \frac{\ln(2)}{m} \leq \delta.$$

$\square$

## 4 Second Moment of the Estimation Error

In this section we prove eq. (5) of Theorem 1.2. It will be more convenient to directly work with the unbiased version of $M$. Specifically, we define $L(S,z) \doteq M(S,z) - \mathcal{E}_{\mathcal{P}}[M(S)]$. Clearly, $L$ is *unbiased* with respect to $\mathcal{P}$ in the sense that for every $S \in Z^n$, $\mathcal{E}_{\mathcal{P}}[L(S)] = 0$. Note that if the range of $M$ is $[0,1]$ then the range of $L$ is $[-1,1]$. Further, $L$ has uniform stability of at most $2\gamma$ since for two datasets $S$ and $S'$ that differ in a single element,

$$|\mathcal{E}_{\mathcal{P}}[M(S)] - \mathcal{E}_{\mathcal{P}}[M(S')]| \leq \left|\operatorname*{\mathbf{E}}_{z\sim\mathcal{P}}[M(S,z) - M(S',z)]\right| \leq \gamma.$$

Observe that

$$\Delta_{\mathcal{P}-S}(M(S)) = \frac{1}{n}\sum_{i=1}^n (\mathcal{E}_{\mathcal{P}}[M(S)] - M(S,S_i)) = \frac{-1}{n}\sum_{i=1}^n L(S,S_i) = -\mathcal{E}_S[L(S)]. \quad (7)$$

By eq. (7) we obtain that

$$\operatorname*{\mathbf{E}}_{S\sim\mathcal{P}^n}\left[(\Delta_{\mathcal{P}-S}(M(S)))^2\right] = \operatorname*{\mathbf{E}}_{S\sim\mathcal{P}^n}\left[(\mathcal{E}_S[L(S)])^2\right].$$

Therefore eq. (5) of Theorem 1.2 will follow immediately from the following lemma (by using it with stability $2\gamma$).

**Lemma 4.1.** *Let* $L\colon Z^n \times Z \to [-1,1]$ *be a data-dependent function with uniform stability $\gamma$ and $\mathcal{P}$ be an arbitrary distribution over $Z$. If $L$ is unbiased with respect to $\mathcal{P}$ then:*

$$\operatorname*{\mathbf{E}}_{S\sim\mathcal{P}^n}\left[(\mathcal{E}_S[L(S)])^2\right] \leq 4\gamma^2 + \frac{2}{n}.$$

Our proof starts by first establishing this result for the leave-one-out estimate.

**Lemma 4.2.** *For a data-dependent function $L\colon Z^n \times Z \to [-1,1]$, a dataset $S \in Z^n$ and a distribution $\mathcal{P}$, define*

$$\mathcal{E}_S\left[L\left(S^{\leftarrow\mathcal{P}}\right)\right] \doteq \operatorname*{\mathbf{E}}_{z\sim\mathcal{P}}\left[\frac{1}{n}\sum_{i\in[n]}L(S^{i\leftarrow z}, S_i)\right].$$

*If $L$ has uniform stability $\gamma$ and is unbiased with respect to $\mathcal{P}$ then:*

$$\operatorname*{\mathbf{E}}_{S\sim\mathcal{P}^n}\left[\left(\mathcal{E}_S\left[L\left(S^{\leftarrow\mathcal{P}}\right)\right]\right)^2\right] \leq \gamma^2 + \frac{1}{n}.$$

*Proof.*

$$\mathop{\mathbf{E}}_{S\sim\mathcal{P}^n}\left[\left(\mathcal{E}_S\left[L\left(S^{\leftarrow\mathcal{P}}\right)\right]\right)^2\right] \leq \mathop{\mathbf{E}}_{S\sim\mathcal{P}^n,z\sim\mathcal{P}}\left[\left(\frac{1}{n}\sum_{i\in[n]}L(S^{i\leftarrow z},S_i)\right)^2\right]$$

$$= \frac{1}{n^2}\sum_{i\in[n]}\mathop{\mathbf{E}}_{S\sim\mathcal{P}^n,z\sim\mathcal{P}}\left[\left(L(S^{i\leftarrow z},S_i)\right)^2\right] + \frac{1}{n^2}\sum_{i,j\in[n],i\neq j}\mathop{\mathbf{E}}_{S\sim\mathcal{P}^n,z\sim\mathcal{P}}\left[L(S^{i\leftarrow z},S_i)\cdot L(S^{j\leftarrow z},S_j)\right]$$

$$\leq \frac{1}{n} + \frac{1}{n^2}\sum_{i,j\in[n],i\neq j}\mathop{\mathbf{E}}_{S\sim\mathcal{P}^n,z\sim\mathcal{P}}\left[L(S^{i\leftarrow z},S_i)\cdot L(S^{j\leftarrow z},S_j)\right], \tag{8}$$

where we used convexity to obtain the first line and the bound on the range of $L$ to obtain the last inequality. For a fixed $i\neq j$ and a fixed setting of all the elements in $S$ with other indices (which we denote by $S^{-i,j}$) we now analyze the cross term

$$v_{i,j} \doteq \mathop{\mathbf{E}}_{S_i,S_j,z\sim\mathcal{P}}\left[L(S^{i\leftarrow z},S_i)\cdot L(S^{j\leftarrow z},S_j)\right].$$

For $z\in Z$, define

$$g(z) = \min_{z_i,z_j\in Z}L(S^{i,j\leftarrow z_i,z_j},z) + \gamma.$$

(We remark that $g$ implicitly depends on $i,j$ and $S^{-i,j}$). Uniform stability of $L$ implies that

$$\max_{z_i,z_j\in Z}L(S^{i,j\leftarrow z_i,z_j},z) \leq \min_{z_i,z_j\in Z}L(S^{i,j\leftarrow z_i,z_j},z) + 2\gamma.$$

This means that for all $z_i,z_j,z\in Z$,

$$\left|L(S^{i,j\leftarrow z_i,z_j},z) - g(z)\right| \leq \gamma. \tag{9}$$

Using this inequality we obtain

$$v_{i,j} = \mathop{\mathbf{E}}_{S_i,S_j,z\sim\mathcal{P}}\left[L(S^{i\leftarrow z},S_i)\cdot L(S^{j\leftarrow z},S_j)\right]$$

$$= \mathop{\mathbf{E}}_{S_i,S_j,z\sim\mathcal{P}}\left[(L(S^{i\leftarrow z},S_i) - g(S_i))\cdot(L(S^{j\leftarrow z},S_j) - g(S_j))\right] + \mathop{\mathbf{E}}_{S_i,S_j,z\sim\mathcal{P}}\left[g(S_i)\cdot L(S^{j\leftarrow z},S_j)\right]$$

$$+ \mathop{\mathbf{E}}_{S_i,S_j,z\sim\mathcal{P}}\left[g(S_j)\cdot L(S^{i\leftarrow z},S_i)\right] - \mathop{\mathbf{E}}_{S_i,S_j\sim\mathcal{P}}\left[g(S_i)\cdot g(S_j)\right]$$

$$\leq \gamma^2 + \mathop{\mathbf{E}}_{S_i,S_j,z\sim\mathcal{P}}\left[g(S_i)\cdot L(S^{j\leftarrow z},S_j)\right] + \mathop{\mathbf{E}}_{S_i,S_j,z\sim\mathcal{P}}\left[g(S_j)\cdot L(S^{i\leftarrow z},S_i)\right] - \left(\mathop{\mathbf{E}}_{z'\sim\mathcal{P}}[g(z')]\right)^2.$$

Note that $L$ is unbiased and $g$ does not depend on $S_i$ or $S_j$. Therefore, for every fixed setting of $S_i$ and $z$,

$$\mathop{\mathbf{E}}_{S_j\sim\mathcal{P}}\left[g(S_i)\cdot L(S^{j\leftarrow z},S_j)\right] = g(S_i)\cdot\mathcal{E}_\mathcal{P}[L(S^{j\leftarrow z})] = 0.$$

Therefore,

$$\mathop{\mathbf{E}}_{S_i,S_j,z\sim\mathcal{P}}\left[g(S_i)\cdot L(S^{j\leftarrow z},S_j)\right] + \mathop{\mathbf{E}}_{S_i,S_j,z\sim\mathcal{P}}\left[g(S_j)\cdot L(S^{i\leftarrow z},S_i)]\right] = 0.$$

implying that $v_{i,j}\leq\gamma^2$. Substituting this into eq.(8) we obtain the claim. $\qquad\square$

We can now obtain the proof of Lemma 4.1 by observing that for every $S$, the empirical mean $\mathcal{E}_S[L(S)]$ is within $\gamma$ of our leave-one-out estimator $\mathcal{E}_S\left[L\left(S^{\leftarrow\mathcal{P}}\right)\right]$(see supplemental material for the proof).

## Footnotes

[1]Also referred to as the *generalization gap* is several recent works.

[2]The analysis in [14] focuses on the stochastic gradient descent and derives uniform stability for the expectation of the loss (over the randomness of the algorithm). However their analysis applies to gradient steps on smooth functions more generally.

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
