[Supplementary Material]

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

*Proof of Lemma 4.1.* Observe that the uniform stability of $L$ implies that for every $S$,

$$\left|\mathcal{E}_S[L(S)] - \mathcal{E}_S\left[L\left(S^{\leftarrow\mathcal{P}}\right)\right]\right| = \left|\frac{1}{n}\sum_{i\in[n]}L(S,S_i) - \underset{z\sim\mathcal{P}}{\mathbf{E}}\left[\frac{1}{n}\sum_{i\in[n]}L(S^{i\leftarrow z},S_i)\right]\right|$$

$$\leq \frac{1}{n}\sum_{i\in[n]}\underset{z\sim\mathcal{P}}{\mathbf{E}}\left[\left|L(S,S_i) - L(S^{i\leftarrow z},S_i)\right|\right] \leq \gamma. \tag{10}$$

Hence

$$\underset{S\sim\mathcal{P}^n}{\mathbf{E}}\left[\left(\mathcal{E}_S[L(S)]\right)^2\right] = \underset{S\sim\mathcal{P}^n}{\mathbf{E}}\left[\left(\mathcal{E}_S\left[L\left(S^{\leftarrow\mathcal{P}}\right)\right] + \mathcal{E}_S[L(S)] - \mathcal{E}_S\left[L\left(S^{\leftarrow\mathcal{P}}\right)\right]\right)^2\right]$$

$$\leq 2\cdot\underset{S\sim\mathcal{P}^n}{\mathbf{E}}\left[\left(\mathcal{E}_S\left[L\left(S^{\leftarrow\mathcal{P}}\right)\right]\right)^2\right] + 2\cdot\underset{S\sim\mathcal{P}^n}{\mathbf{E}}\left[\left(\mathcal{E}_S[L(S)] - \mathcal{E}_S\left[L\left(S^{\leftarrow\mathcal{P}}\right)\right]\right)^2\right]$$

$$\leq 2\left(\gamma^2 + \frac{1}{n}\right) + 2\gamma^2 = 4\gamma^2 + \frac{2}{n}.$$

where we used the Cauchy-Schwarz to obtain the second line and Lemma 4.2 together with eq. (10) to obtain the third line. $\qquad\square$

## 5  Applications

We now apply our bounds on the estimation error to several known uniformly stable algorithms. Many additional applications can be derived in a similar manner.

### 5.1  Learning via Stochastic Convex Optimization

We consider learning problems that can be formulated as stochastic convex optimization. Specifically, these are problems in which the goal is to minimize the expected loss:

$$F_\mathcal{P}(w) \doteq \underset{z\sim\mathcal{P}}{\mathbf{E}}[\ell(w, z)],$$

over $w \in \mathcal{K} \subset \mathbb{R}^d$ for some convex body $\mathcal{K}$ and a family of convex losses $\mathcal{F} = \{\ell(\cdot, z)\}_{z\in Z}$. The stochastic convex optimization problem for $\mathcal{F}$ is the problem of minimizing $F_\mathcal{P}(w)$ over $\mathcal{K}$ for an arbitrary distributions $\mathcal{P}$ over $Z$.

Many learning problems can be expressed in or relaxed to this general form. As a result many optimization algorithms are known and the optimal error rates are understood for a variety of families of convex functions. However most of these results are obtained via algorithm-specific techniques such as online-to-batch conversion [5] and stability-based arguments rather than uniform convergence. As it turns out, this is unavoidable. This was first pointed out in the seminal work of Shalev-Shwartz et al. [26] who showed that there is exists a gap between the bounds that can be obtained via uniform convergence (or ERM algorithms) and bounds achievable via alternative approaches.

For concreteness, let $\mathcal{F}$ be the family of all convex 1-Lipschitz losses over the unit Euclidean ball in $d$ dimension (denoted by $\mathcal{B}_2^d(1)$). It is well-known that in this case the stochastic convex optimization problem can be solved with error $1/\sqrt{n}$ via projected SGD. At the same time it was shown in [26] that there exists an algorithm that minimizes the empirical error while having the worst case error of $\Omega\left(\frac{\log d}{n}\right)$. This has been subsequently strengthened to $\Omega\left(\frac{d}{n}\right)$ by Feldman [13] who also showed a lower bound of $\Omega\left(\sqrt{\frac{d}{n}}\right)$ for obtaining uniform convergence in this setting. Further, with Lipschitzness assumption replaced by the assumption that functions have range in $[0, 1]$ the gap becomes infinite even for $d = 2$ [13].

**Strongly convex ERM**  We now revisit the stability results known for this basic setting [4, 26] (for simplicity and without loss of generality we will scale the domain and functions to 1).

**Theorem 5.1** ([26]). *Let $\mathcal{K} \subseteq \mathcal{B}_2^d(1)$ be a convex body, $\mathcal{F} = \{\ell(\cdot, z) \mid z \in Z\}$ be a family of 1-Lipschitz, $\lambda$-strongly convex loss functions over $\mathcal{K}$ with range in $[0, 1]$. For a dataset $S \in Z^n$ let $w_S$ denote the empirical minimizer of loss on S: $w_S = \operatorname{argmin}_{w\in\mathcal{K}} \sum_{i\in[n]} \ell(w, S_i)$. Then the algorithm that given S outputs $w_S$ has uniform stability $\frac{4}{\lambda n}$ with respect to loss $\ell$. Further, for every distribution $\mathcal{P}$ over $Z$ and $\delta > 0$:*

$$\underset{S\sim\mathcal{P}^n}{\mathbf{Pr}}\left[F_\mathcal{P}(w_S) \geq \min_{w\in\mathcal{K}} F_\mathcal{P}(w) + \frac{4}{\delta\lambda n}\right] \leq \delta.$$

We note that the bound on estimation error is obtained by applying Markov's inequality to eq. (4). Theorem 5.1 requires strong convexity. As pointed out in [26], it is possible to add a strongly convex regularizing term $\frac{\lambda}{2}\|w\|^2$ to the objective function that has sufficiently small effect on the loss

function while ensuring stability (and generalization). Specifically, by setting $\lambda = \frac{4}{\sqrt{\delta n}}$ the objective function will change by at most $\lambda$ since $w$ is assumed to be in a ball of radius 1. Plugging this value of $\lambda$ into Thm. 5.1 and accounting for the additional error they get:

**Corollary 5.2** ([26]). *Let $\mathcal{K} \subseteq \mathcal{B}_2^d(1)$ be a convex body, $\mathcal{F} = \{\ell(\cdot, z) \mid z \in Z\}$ be a family of convex 1-Lipschitz loss functions over $\mathcal{K}$ with range in $[0, 1]$. For a dataset $S \in Z^n$ let $w_S$ denote the empirical minimizer of regularized loss on $S$: $w_{S,\lambda} = \mathrm{argmin}_{w \in \mathcal{K}} \sum_{i \in [n]} \ell(w, S_i) + \frac{\lambda}{2} \|w\|_2^2$. For every distribution $\mathcal{P}$ over $Z$ and $\delta > 0$ using $\lambda = \frac{4}{\sqrt{\delta n}}$ gives:*

$$\Pr_{S \sim \mathcal{P}^n} \left[ F_{\mathcal{P}}(w_{S,\lambda}) \geq \min_{w \in \mathcal{K}} F_{\mathcal{P}}(w) + \frac{4}{\sqrt{\delta n}} \cdot \left( 1 + \frac{8}{\delta n} \right) \right] \leq \delta.$$

We now spell out the results for these settings implied by our generalization bounds.

**Corollary 5.3.** *In the setting of Theorem 5.1, for some fixed constants $c_1$ and $c_2$:*

$$\Pr_{S \sim \mathcal{P}^n} \left[ F_{\mathcal{P}}(w_S) \geq \min_{w \in \mathcal{K}} F_{\mathcal{P}}(w) + c_1 \left( \frac{1}{\sqrt{\delta \lambda n}} + \frac{1}{\sqrt{n}} \right) \right] \leq \delta,$$

*and*

$$\Pr_{S \sim \mathcal{P}^n} \left[ F_{\mathcal{P}}(w_S) \geq \min_{w \in \mathcal{K}} F_{\mathcal{P}}(w) + \frac{c_2 \sqrt{\ln(1/\delta)}}{\sqrt{\lambda n}} \right] \leq \delta.$$

The first part of this corollary follows directly from applying Chebyshev's inequality to eq. (5) in Theorem 1.2. To apply our results in the setting of Corollary 5.2 we will use a different choice of $\lambda$ to minimize the error. Specifically, we will choose $\lambda = c/\sqrt{\sqrt{\delta} n}$ for some constant $c$ when using the second moment and $\lambda = c/n^{2/3}$ when using the high probability result.

**Corollary 5.4.** *In the setting of Corollary 5.2 with appropriate choices of $\lambda$ and some fixed constants $c_1$ and $c_2$:*

$$\Pr_{S \sim \mathcal{P}^n} \left[ F_{\mathcal{P}}(w_{S,\lambda}) \geq \min_{w \in \mathcal{K}} F_{\mathcal{P}}(w) + \frac{c_1}{\delta^{1/4} \sqrt{n}} \right] \leq \delta,$$

*and*

$$\Pr_{S \sim \mathcal{P}^n} \left[ F_{\mathcal{P}}(w_{S,\lambda}) \geq \min_{w \in \mathcal{K}} F_{\mathcal{P}}(w) + \frac{c_2 \sqrt{\ln(1/\delta)}}{n^{1/3}} \right] \leq \delta.$$

**Gradient descent on smooth functions**   We now recall the results of Hardt et al. [14] for convex and smooth functions. These results derive their guarantees from the fact that a gradient step on a sufficiently smooth loss function is non-expansive. That is, for any pair of points $w$ and $w'$, any $\beta$-smooth convex function $f$, and $0 \leq \eta \leq 2/\beta$,

$$\|(w - \eta \nabla f(w)) - (w' - \eta \nabla f(w'))\| \leq \|w - w'\|.$$

Projection to a convex body is also non-expansive. This implies that the effect of each datapoint $S_i$ on the loss of the solution can be bounded by $\sum_t \eta_{t,i} \|\nabla \ell(w_t, S_i)\|$, where $\eta_{t,i}$ is the rate with which point $S_i$ is used at step $t$. Hence this analysis can be used for a variety of versions of gradient descent with different rates, arbitrary batch sizes and multiple passes over the data. For most of such algorithms no alternative analyses of estimation error are known. It also means that the estimation error can be bounded without any assumptions on how close the output of the algorithm to the empirical minimum.

For concreteness, we apply the bounds from [14] to projected gradient descent on the empirical objective. Unlike for single-pass algorithms, we are not aware of any other approaches to proving generalization guarantees for this algorithm. For an integer $T$, and dataset $S$, let $\mathrm{PGD}_T(S)$ denote the output of the algorithm that starting from $w_0$ being the origin, performs the following iterative updates for every $t \in [T]$:

$$w_{t+1} \leftarrow \mathrm{Project}_{\mathcal{K}} \left( w_t + \frac{1}{\sqrt{T}} \nabla F_S(w_t) \right),$$

where $F_S(w)$ is the empirical objective function $\frac{1}{n} \sum_{i=1}^{n} \ell(w, S_i)$ and $\mathrm{Project}_{\mathcal{K}}$ denotes projection to $\mathcal{K}$. The algorithm returns the average iterate: $\bar{w}_S \doteq \frac{1}{T} \sum_{t \in [T]} w_t$.

**Theorem 5.5** ([14]). *Let $\mathcal{K} \subseteq \mathcal{B}_2^d(1)$ be a convex body, $\mathcal{F} = \{\ell(\cdot, z) \mid z \in Z\}$ be a family of convex 1-Lipschitz and $\sigma$-smooth loss functions over $\mathcal{K}$ with range in $[0, 1]$. For an integer $T$ and a dataset $S \in Z^n$, let $\bar{w}_{S,T} = PGD_T(S)$. If $\sigma \leq 2/\sqrt{T}$ then $PGD_T(S)$ has uniform stability $\sqrt{T}/n$ with respect to loss $\ell$. Further,*

$$F_S(\bar{w}_{S,T}) \leq \min_{w \in \mathcal{K}} F_S(w) + \frac{2}{\sqrt{T}}.$$

*and for every distribution $\mathcal{P}$ over $Z$:*

$$\operatorname*{\mathbf{E}}_{S \sim \mathcal{P}^n} [F_{\mathcal{P}}(\bar{w}_{S,T})] \leq \min_{w \in \mathcal{K}} F_{\mathcal{P}}(w) + \frac{2}{\sqrt{T}} + \frac{\sqrt{T}}{n}.$$

To minimize the expected true loss the algorithm needs to be used with $T = n/\sqrt{2}$, which implies that the stability parameter is $\Omega(1/\sqrt{n})$. We remark that in this case even "low-probability" generalization results cannot be obtained directly from the bound on the expectation of the true loss.

Applying eq. (5) with Chebyshev's inequality to the results of Theorem 5.5 gives that for some constant $c_1$ and every $\delta > 0$:

$$\operatorname*{\mathbf{Pr}}_{S \sim \mathcal{P}^n} \left[ F_{\mathcal{P}}(\bar{w}_{S,T}) \geq \min_{w \in \mathcal{K}} F_{\mathcal{P}}(w) + \frac{2}{\sqrt{T}} + \frac{c_1}{\sqrt{\delta}} \left( \frac{\sqrt{T}}{n} + \frac{1}{\sqrt{n}} \right) \right] \leq \delta.$$

At the same time eq. (6) gives (for some constant $c_2$):

$$\operatorname*{\mathbf{Pr}}_{S \sim \mathcal{P}^n} \left[ F_{\mathcal{P}}(\bar{w}_{S,T}) \geq \min_{w \in \mathcal{K}} F_{\mathcal{P}}(w) + \frac{2}{\sqrt{T}} + \frac{c_2 T^{1/4} \sqrt{\log(1/\delta)}}{\sqrt{n}} \right] \leq \delta.$$

By optimizing the choice of $T$ we can get essentially the same rates as we have obtained for the ERM in Corollary 5.4 (although in this case we need a smoothness assumption).

**Corollary 5.6.** *In the setting of Theorem 5.5 with appropriate choices of $T$, for every distribution $\mathcal{P}$ over $Z$, $\delta > 0$, some fixed constants $c_1$ and $c_2$:*

$$\operatorname*{\mathbf{Pr}}_{S \sim \mathcal{P}^n} \left[ F_{\mathcal{P}}(\bar{w}_{S,T}) \geq \min_{w \in \mathcal{K}} F_{\mathcal{P}}(w) + \frac{c_1}{\delta^{1/4}\sqrt{n}} \right] \leq \delta,$$

*and*

$$\operatorname*{\mathbf{Pr}}_{S \sim \mathcal{P}^n} \left[ F_{\mathcal{P}}(\bar{w}_{S,T}) \geq \min_{w \in \mathcal{K}} F_{\mathcal{P}}(w) + \frac{c_2 \sqrt{\ln(1/\delta)}}{n^{1/3}} \right] \leq \delta.$$

### 5.2 Privacy-Preserving Prediction

Our results can also be used to improve the bounds on generalization error of learning algorithms with differentially private prediction. These are algorithms introduced to model privacy-preserving learning in the settings where users only have black-box access to the model via a prediction interface [10]. Formally,

**Definition 5.7** ([10]). *Let $K$ be an algorithm that given a dataset $S \in (X \times Y)^n$ and a point $x \in X$ produces a value in $Y$. Then $K$ is $\epsilon$-differentially private prediction algorithm if for every $x \in X$, the output $K(S, x)$ is $\epsilon$-differentially private with respect to $S$.*

The properties of differential privacy imply that the expectation over the randomness of $K$ of the loss of $K$ at any point is uniformly stable. Specifically, for every $\epsilon$-differentially private prediction algorithm, every loss function $\ell : Y \times Y \to [0, 1]$, two datasets $S$ and $S'$ that differ in a single element and $(x, y) \in X \times Y$ we have that

$$\operatorname*{\mathbf{E}}_K [\ell(K(S, x), y)] \leq e^\epsilon \cdot \operatorname*{\mathbf{E}}_K [\ell(K(S', x), y)].$$

In particular, this implies that

$$\left| \operatorname*{\mathbf{E}}_K [\ell(K(S, x), y)] - \operatorname*{\mathbf{E}}_K [\ell(K(S', x), y)] \right| \leq e^\epsilon - 1.$$

Therefore our generalization bounds can be applied to the data-dependent function $\mathbf{E}_K[\ell(K(S, x), y)]$. This gives the following corollary of Theorem 1.2:

**Theorem 5.8.** *Let $K : (X \times Y)^n \times X \to Y$ be an $\epsilon$-differentially private prediction and $\ell \colon Y \times Y \to [0,1]$ be an arbitrary loss function. For a probability distribution $\mathcal{P}$ over $Z$ we define:*

$$\Delta_{\mathcal{P}-S}(\mathbf{E}[\ell(K)]) \doteq \underset{(x,y)\sim\mathcal{P},K}{\mathbf{E}}[\ell(K(S,x),y)] - \frac{1}{n}\sum_{i=1}^{n} \underset{K}{\mathbf{E}}[\ell(K(S,x_i),y_i)].$$

*Then for any $\delta \in (0,1)$:*

$$\underset{S\sim\mathcal{P}^n}{\mathbf{E}}\left[(\Delta_{\mathcal{P}-S}(\mathbf{E}[\ell(K)]))^2\right] \leq 16(e^\epsilon - 1)^2 + \frac{2}{n};$$

$$\underset{S\sim\mathcal{P}^n}{\mathbf{Pr}}\left[\Delta_{\mathcal{P}-S}(\mathbf{E}[\ell(K)]) \geq 8\sqrt{\left(2(e^\epsilon - 1) + \frac{1}{n}\right)\cdot \ln(8/\delta)}\right] \leq \delta.$$

These bounds are stronger than those obtained in [10] in several parameter regimes (but are more generally incomparable since bounds in [10] are multiplicative).

Dwork and Feldman [10] describe an algorithm for agnostically learning threshold functions on a line with differentially private prediction. They demonstrate that their algorithm achieves low empirical error. The complexity of models that their algorithm produces is unbounded and therefore the estimation error cannot be bounded via uniform convergence. Hence they appeal to generalization properties of differentially private prediction. Theorem 5.8 directly implies stronger generalization bounds for this algorithm (we omit more formal details since they require several additional definitions and the application itself is straightforward).

## Footnotes

[1] Also referred to as the *generalization gap* is several recent works.

[2]The analysis in [14] focuses on the stochastic gradient descent and derives uniform stability for the expectation of the loss (over the randomness of the algorithm). However their analysis applies to gradient steps on smooth functions more generally.