[Reviews · NeurIPS 2018]

Reviewer 1



The paper presents improved high probability bounds on the generalization error for uniformly stable algorithms. The new bound is of order sqrt(gamma + 1/n) for the gamma-uniformly-stable algorithms, as opposed to the well-known (gamma + 1/n) * sqrt(n) of Bousquet and Elisseeff. The first consequence is that one obtains non-vacuous bounds for algorithms even when gamma = O(1/n^a) for some a > 0, as opposed to a >= 1 in previous bounds. This allows to show high-probability bounds for not necessarily strongly convex objectives. For example, as authors also note, Hardt et al. show that SGD on convex and smooth problems is 1/sqrt(n)-stable. The second consequence is that for regularized ERM one gets the correct shape (obtainable through uniform convergence type bounds) w.r.t. regularization parameter (lambda), i.e. 1/sqrt(n*lambda), which improves upon 1/(sqrt(n)*lambda) of Bousquet and Elisseeff. This was, however, also observed in prior works on stability. On a disappointing side, authors did not sufficiently discuss recent works in the literature on stability (and did not properly position their results), which obtain similar bounds albeit in slightly more restrictive settings. For example, in [18] authors derive similarly shaped bounds for algorithms that output hypotheses in Banach spaces Another result is by A. Maurer [*1] which analyses regularized convex ERM. Technically, the paper shows these bounds by exploiting proof techniques used in differential privacy, and (rather compact) analysis might be of interest on its own. On this note, one question to authors: how easily this can be extended to weaker notions of stability, i.e. distribution-dependent ones? What are possible challenges? In addition, the paper is well-written and presentation is well-done. [*1] A. Maurer "A Second-order Look at Stability and Generalization" (COLT 2017)

Reviewer 2



This paper develops new generalization bounds for algorithms that are uniformly stable. Specifically, the main results are an improved high probability bound on generalization error and an improved bound on the second moment on the generalization error. Both bounds to appear to be real improvements over past results and advance what is known about the generalization error of uniformly stable algorithms. The key tools used to prove the new results are from recent developments in differential privacy and adaptive data analysis. While many of the core techniques were established in previous works, the authors do go further to combine these techniques with a more learning-specific analysis to get their results, and I believe their supporting lemmas and proofs of their theorems represent non-trivial contributions. An unfortunate weakness of this paper is that it provides only a blurry coverage of applications near the end of the Introduction, but all new results related to applications are hidden away in the appendix. New results do not belong in the appendix, and to expect reviewers to take them into account is to implicitly try to overcome the page limit. Therefore, with regards to applications, my review only considers whatever I could learn from the very high-level Applications section on page 4. I recommend the authors remove Appendix A from the paper, and that they try to develop at least one of the applications properly in the main text. DETAILED REVIEW In comparing the results of Theorem 1.2 to what was previously known from equations (2) and (3), I found this paper's results to be significant improvements. These results are foundational and I expect can be further built upon to get new results in specific (still theoretical) applications. On the whole, the paper is clear, but I recommend rewriting the Applications subsection as it was hard to get a clear view into the authors' results here as so much is packed in within a small space. A good tradeoff would be to move some of the proofs appearing in the main text into the appendix, allowing more space in the main text for formal statements of (some of the) Applications-related results. This will no doubt improve the impact of the paper as more people will see what the Theorem 1.2 can buy us in practice''. Also, the citations for Theorem 3.3 were not very clear. Some more guidance as to what parts of these papers have the stated result would help (I found them finally but after a lot of extra effort on my part). Regarding the technical quality of the authors' results: The new technical results of the authors are Theorem 1.2 (consisting of inequalities (5) and (6)) and its supporting lemmas: Lemma 3.4, 3.5 ,and 4.1, and 4.2. I looked through the proofs of all these results. I believe the proofs of the lemmas and also inequaity (5) of Theorem 1.2 are all correct. The proof of inequality (6) of Theorem 1.2 also is generally fine, modulo some small typos / missing details that I ask the authors to include. Specifically: 1. On the 4th line of page 7, when you say By Lemma 3.5 we have that'' I think you should also mention that you are using the part of Theorem 3.3 that says that the exponential mechanism in this case is differentially private, as this is needed to apply Lemma 3.5. 2. When you say To bound this expression we choose $\epsilon =$ ...'', the second equality is only an inequality (it should be $\leq$), and only if you set $\epsilon$ to the RHS (the term with the $\ln(e \ln(2)/\delta)$) does the rest of the proof work. I suggest making this fix so that the proof is correct. In the second to last math display (in Proof of eq. (6) in Theorem 1.2), I wasn't able to see immediately how the inequality follows (even after using $\gamma \leq \sqrt{\gamma}$). Saying a little extra here could help. 3. I was able to myself construct the argument for why you need only consider the regime of $(e^\epsilon - 1) \leq 2 \epsilon$, but as stated, your explanation of Our bound is at least $2 \epsilon$ and hence...'' is not very clear and should be rephrased/expanded. Last, while many of the core original tools were developed in previous works, I believe that some amount of ingenuity was needed on the parts of the authors to apply these tools in the right way to get their new results for uniformly stable algorithms. As these techniques are still relatively new to the learning theory community, I believe mixing them into this space provides significant value. Minor corrections/points: In Definition 2.2, I believe $S^{(i \leftarrow z_i)}$ should be replaced by $S^{(i \leftarrow z)}$ There are suspicious issues with missing line numbers thorughout the paper, but especially on page 7. EXPERTISE OF REVIEWER I am familiar with the fundamental developments related to algorithmic stability (and more generally, learning theory), although I am less aware of new contributions within the last year or so. Likewise, I am familiar with many of the techniques used in differential privacy and adaptive data analysis. UPDATE AFTER READING THE REBUTTAL You should reconsider leaving the current Applications section as is. I think it really is not useful and will hurt the impact of your paper. It is sad that you would leave a discussion of core selling points of your paper to the appendix. While a learning theorist may get excited, a large number of other readers will not. Also, things that "follow easily" to the authors may not follow so easily to a reader that is not an author of the work; please consider the perspective the reader.

Reviewer 3



The paper establishes two improved generalization bounds for uniformly stable algorithms. Using the approach introduced in [2,21] for differentially private algorithms, the authors establish 1) an upper bound on the second moment of the generalization error that matches the lower bound on the first moment (up to constants), and 2) an upper bound on the tail of the generalization error that improves previously known results (the tightness of this results is left for further investigation). The results are shown to imply stronger guarantees for well-studied algorithms, specifically in the common situation when the stability parameter scales with the inverse of the square-root of the sample size. Examples are developed in the supplementary materials and include 1) ERM for 1-Lipschitz convex functions bounded in [0,1], improving the low-probability results in [25] and establishing high-probability bounds; 2) gradient descent on smooth convex functions, establishing low- and high- probability bounds for the setting explored in [14]; 3) learning algorithms with differentially private prediction, establishing bounds that are stronger than the (multiplicative) bounds established in [10] in some parametric regimes. The paper is well-written, with the main contributions clearly emphasized and commented. Connection with previous literature is detailed, and the proofs are well-commented and easy to follow. The elements of novelty that add to the approach in [2,21] are also well-described (see Lemma 3.5, for instance). The paper offers new insights on algorithmic stability, which are particularly relevant to the field of machine learning these days, given the key role that stability has played in the analysis of widely-used algorithms (currently, stability seems the only tool to understand the generalization properties of multiple-pass stochastic gradient descent, as in [14]) and given its connection with differential privacy, another main topic of research in the field.